

# Drainage of an ice-dammed lake through a supraglacial stream: hydraulics and thermodynamics

Christophe Ogier[1,2], Mauro A. Werder[1,2], Matthias Huss[1,2,3], Isabelle Kull[4], David Hodel[5], and Daniel Farinotti[1,2]

[1]Laboratory of Hydraulics, Hydrology and Glaciology (VAW), ETH Zurich, Zurich, Switzerland
[2]Swiss Federal Institute for Forest, Snow and Landscape Research (WSL), Birmensdorf, Switzerland
[3]Department of Geosciences, University of Fribourg, Fribourg, Switzerland
[4]Geotest AG, Zollikofen, Switzerland
[5]Theiler Ingenieure, Thun, Switzerland

**Correspondence:** Christophe Ogier (ogier@vaw.baug.ethz.ch)

## 1 Abstract

The glacier-dammed Lac des Faverges, located on Glacier de la Plaine Morte (Swiss Alps), drained annually as a glacier lake outburst flood since 2011. In 2018, the lake volume reached more than $2 \times 10^6 \, \text{m}^3$ and the resulting flood caused damages to the infrastructure downstream. In 2019, a supraglacial channel was dug to artificially initiate a surface lake drainage, thus lim-

iting the lake water volume and the corresponding hazard. The peak in lake discharge was successfully reduced by over 90 % compared to 2018. We conducted extensive field measurements of the lake-channel system during the 48-days drainage event of 2019 to characterize its hydraulics and thermodynamics. The derived Darcy-Weisbach friction factor, which characterizes the water flow resistance in the channel, ranges from 0.17 to 0.48. This broad range emphasizes the factor's variability, and questions the choice of a constant friction factor in glacio-hydrological models. For the Nusselt number, which relates the

channel-wall melt to the water temperature, we show that the classic, empirical Dittus-Boelter equation with the standard coefficients is not adequately representing our measurements, and we propose a suitable pair of coefficients to fit our observations. This hints at the need to continue the research into how heat transfer at the ice/water interface is described in the context of glacial hydraulics.

## 2 Introduction

Glacier-dammed lakes are often unstable as ice dams are prone to rapidly fail which leads to partial or total drainage of the impounded lake through supraglacial, englacial and subglacial conduits (Roberts, 2005). The sudden release of the water impacts glacier dynamics (Röthlisberger, 1972) and may lead to extreme peak discharge at the outlet (Björnsson, 1992). Lake dam failure can occur via three main mechanisms, or a combination thereof, which are the following: (i) high water pressure beneath the dam leads to its flotation (Björnsson, 2010), (ii) the lake water leaks through the dam via e.g. pre-existing cracks,

channels form and then progressively enlarge (Nye, 1976), or (iii) the lake water overspills the dam and forms a breach due to





ice erosion (Walder and Costa, 1996; Raymond and Nolan, 2000). The last process is less well-documented than the former two, and is more common for cold-based glaciers rather than temperate ones (Björnsson, 2010). Enlargement of pre-existing cracks and conduits (process ii) is possible due to frictional heating (i.e thermal energy dissipation in the waterflow due to potential energy release) and/or due to sensible heat fluxes (i.e advection of warm water from the lake). In general, both

processes (ii) and (iii) lead to progressively rising discharge, whereas process (i) often results in a very fast drainage onset and high discharge. These fast lake drainages, so-called glacial lake outburst floods (GLOFs) or 'jokulhlaups', are a serious threat in populated areas and have caused major destruction in the past (e.g. Haeberli, 1983; Richardson and Reynolds, 2000; Björnsson, 2002; Ancey et al., 2019).

In the frame of hazard mitigation, glacier-dammed lakes have sometimes been drained artificially. In 1892, for example, an

outburst event at Glacier de Tête Rousse (France) devastated the village of Saint Gervais les Bains and caused 175 fatalities. To prevent further hazardous events, a tunnel in rock and ice was dug in 1904 to empty the subglacial lake (Vincent et al., 2010b). This tunnel has been maintained until today but water did no longer run through it. In 2010, a subglacial water-filled cavity of $55,000\,\mathrm{m^3}$ was discovered at the same glacier through geophysical surveys, and was artificially drained using submersible pumps (Vincent et al., 2012). In some other cases, a channel was dug inside the ice or at the glacier surface to evacuate the lake

water. The earliest of such examples, is from Glacier de Giétro (Switzerland) in 1818, when a channel was dug through the ice dam to empty a lake (maximum volume of about $25 \times 10^6\,\mathrm{m^3}$) impounded by an advancing glacier. Due to high channel erosion, however, large parts of the ice dam collapsed, releasing the remaining water in very short time, leading to 40 fatalities. The discharge peak reconstructed by Ancey et al. (2019) was about $14,500\,\mathrm{m^3s^{-1}}$. In 2005, the $0.7\times 10^6\,\mathrm{m^3}$ ice-dammed lake on Glacier de Rochemelon (French Alps) were also drained artificially. This was done by combining a siphon method and a

surface channel of $100\,\mathrm{m}$ length (Vincent et al., 2010). The dangerous lake was emptied with success, with a peak discharge of merely $1.5\,\mathrm{m^3s^{-1}}$.

In the present paper we focus on glacier lake drainage through a surface channel. When it comes to this sort of intervention for hazard mitigation, it is vital to know whether the drainage will be stable or unstable, i.e. whether the discharge will rise rapidly or not. Raymond and Nolan (2000) introduced the concept of stable and unstable drainage based on observations from

Black Rapids Glacier (Alaska) and identified a set of parameters that are of particular interest. In a stable drainage regime, for example, the lowering rate of the lake level is higher or equal to the channel incision rate and, thus, the lake discharge decreases with time. Conversely, in an unstable drainage regime the channel erosion is higher than the lake level lowering. The lake discharge hence increases with time and the lake is emptied completely and rapidly. Vincent et al. (2010) used the Raymond and Nolan (2000) approach to reconstruct the drainage of the ice-marginal Lac de Rochemelon. They based their

analysis on extensive field measurements carried out during the artificial drainage. They were able to conduct a sensitivity analysis on the relevant parameters that control the lake discharge, such as water temperature and lake area. However, some of their parameters were only inferred *at post* from the field observations, and the analysis was thus not able to predict the peak discharge in advance.

Since then, other studies have tried to model channelized surface drainage in order to focus on the physical processes at

play. Jarosch and Gudmundsson (2012), for example, provided explicit numerical simulations of such drainage by including



the effects of ice dynamics to the pre-existent open-channel flow models. This enabled the shape and evolution of the channel to be purely driven by physics, and not to be pre-defined as in earlier studies (e.g. Raymond and Nolan, 2000; Walder and Costa, 1996). Channel slope, water flux and temperature were shown to be the main parameters controlling channel incision, which in turn dictates the discharge at the lake outlet. Kingslake et al. (2015) built upon the work of Raymond and Nolan
(2000), and formulated a more generally applicable model by including considerations of sub-critical flow at the lake outlet. Although these studies represent the state-of-the-art in supraglacial lake drainage modelling, they have never been validated against independent field observations (Pitcher and Smith, 2019). This calls for corresponding datasets to be acquired, as the question about whether such models are able to correctly simulate supraglacial lake drainage in the context of hazard mitigation remains open.

In this paper, we focus on the collection and interpretation of such a dataset, acquired for the hazardous Lac des Faverges at Glacier de la Plaine Morte (Switzerland). This ice-marginal lake drained subglacially every summer from 2011 to 2018 with increasing volume and peak discharge over time (Huss et al., 2013; Lindner et al., 2020). A monitoring and early warning system was set up in 2012. The drainage event of 2018 caused inundations in the village of Lenk, north of the glacier. Parts of the village needed to be evacuated and damages to houses and infrastructure were substantial. The community thus decided
to design measures to artificially lower the lake level to reduce the hazard potential. In 2019, the lake initially drained through an artificial, supraglacial channel (Figure 1). Later during the summer, half of the lake volume drained subglacially again but without causing damage. We took advantage of this particular situation to carry out extensive field measurement during the 48 days of the lake drainage. In particular, we monitored lake level, discharge, water temperature and channel geometry evolution with high spatial and temporal resolution. This allows us to describe the applied flood risk mitigation strategy in detail, and to
determine some of the most important physical parameters involved in the supraglacial drainage of an ice-dammed lake. We anticipate that this work will support further modelling studies and, thus, also help in the planning of future hazard mitigation measures.

## 3 Study area

### 3.1 Previous GLOFs of Lac des Faverges

Glacier de la Plaine Morte is located in the Bernese Alps (46°23'N, 7°30'E), Switzerland. It is the largest plateau glacier in the European Alps (7.1 km$^2$ in 2019), with 90% of its surface spanning an elevation range of only 2650–2800 m a.s.l. The ice-marginal Lac des Faverges is located in the upper reaches of the glacier, at its south-eastern margin (Figure 1a). According to aerial imagery of the Swiss Federal Office of Topography, the lake started forming in the 1970s, and now fills annually during the melt season. Because of the rapid ice loss over the last years, the basin enlarged, thus increasing the potential lake volume
too (Huss et al., 2013). Simultaneously, the maximum lake level lowered due to a significant reduction in ice surface elevation and since 2012, the lake water no longer overspill a sediment ridge to the south. Instead of draining superficially towards the Rhone basin, the lake water now drains englacially northwards, into the Rhine basin. The glacier lake outburst floods of Lac des Faverges have occurred annually since 2011 (Lindner et al., 2020) and represent a serious concern for Lenk, a 2300-inhabitants





village 10 km downstream of the glacier snout (Bundesamt für Umwelt, 2020) . The lake level and temperature are monitored
in detail since 2012 by Geopraevent AG (https://www.geopraevent.ch/) for early warning purposes, and daily images from an
automatic camera are also available. An alarm is triggered when the rate of lake-level change reaches a given critical value.
Huss et al. (2013) projected the future evolution of Glacier de la Plaine Morte for the coming century, and also estimated the
changes in the lake basin over the next decades. They concluded that a continuous increase in lake volume is likely, along with
an increase in the potential flood hazard for the village of Lenk. In 2018, the lake discharge reached around $80 \, \mathrm{m^3 s^{-1}}$ causing
damage to infrastructure for the first time (Gemeinde Lenk, 2019).

## 3.2   The 2019 GLOF mitigation plan

In spring 2019, local authorities decided to limit the maximum lake volume by constructing a supra- and englacial channel
to artificially drain the lake water in order to face the increasing threat by floods due to sudden lake drainage. This channel
now connects the lake outlet to a permanent large moulin located ~1.3 km westward and about 20m lower in altitude (we will
refer to this feature as to the "Moulin West" in the following, see Figure 1a). Past observations have shown that this moulin
is in turn connected to the subglacial network, and that this connection often establishes relatively early during the melting
season (Finger et al., 2013). In the middle of the channel there is a ~100 m long tunnel (labelled "micro tunnel" in Figure 1c),
which is a remainder of the initial plan to drill a 40 cm-diameter englacial tunnel, instead of a surface channel, for part of the
distance (only a short section of the tunnel was completed, due to technical issues). The supraglacial channel was dug from
the beginning of April until early July 2019. In a first stage, the 4-5 m deep snow cover had to be removed by snowcats. In a
second stage, the ice was cut and removed by an excavator. Because of these artificial interventions, the initial geometry of the
channel is well known, with a width of 1 m at the bottom, 4 m to 7 m depth from the ice surface, and a length of 1.3 km (see
Fig. 1d). During this second stage, water from ice- and snow-melt was present in the channel.

The lake is connected via a preexisting ice-surface canyon and a subsequent natural ice cave to the artificial supraglacial
channel (figure 1b). At the beginning of the canyon, where it connects to the lake, the water flows through an englacial siphon
for about 30 m.

On 10 July 2019, at 11:00, the channel spillway elevation was lowered by an excavator to match the lake level and to
artificially initiate the lake drainage. At that point, the spillway elevation was 2733.15 m a.s.l., corresponding to a lake volume
of $\sim 1.38 \times 10^6 \, \mathrm{m^3}$ and an area of 0.127 km$^2$. The lake water ran only into the first, upper part of the channel (Fig. 1c and d),
and then infiltrated the glacier through a pre-existing moulin located within the micro-tunnel. A dye injection in the channel
on 26 July 2019 revealed that the lake water exited the glacier outlet after about 2 hours, and that there was no significant lake
water accumulation within the glacier.

In the following, we limit our attention to the upper part of the channel through which the lake water flowed for 36 days in
total (Figure 1c and 1d). This section (termed "channel" henceforth), was located between the cave outlet and the micro-tunnel
entrance, was 540 m long and had an average slope of 0.34 %. We designed our field campaign to monitor the hydraulic and
thermodynamic properties of the water flow in the channel, and relate that to lake level and volume evolution.





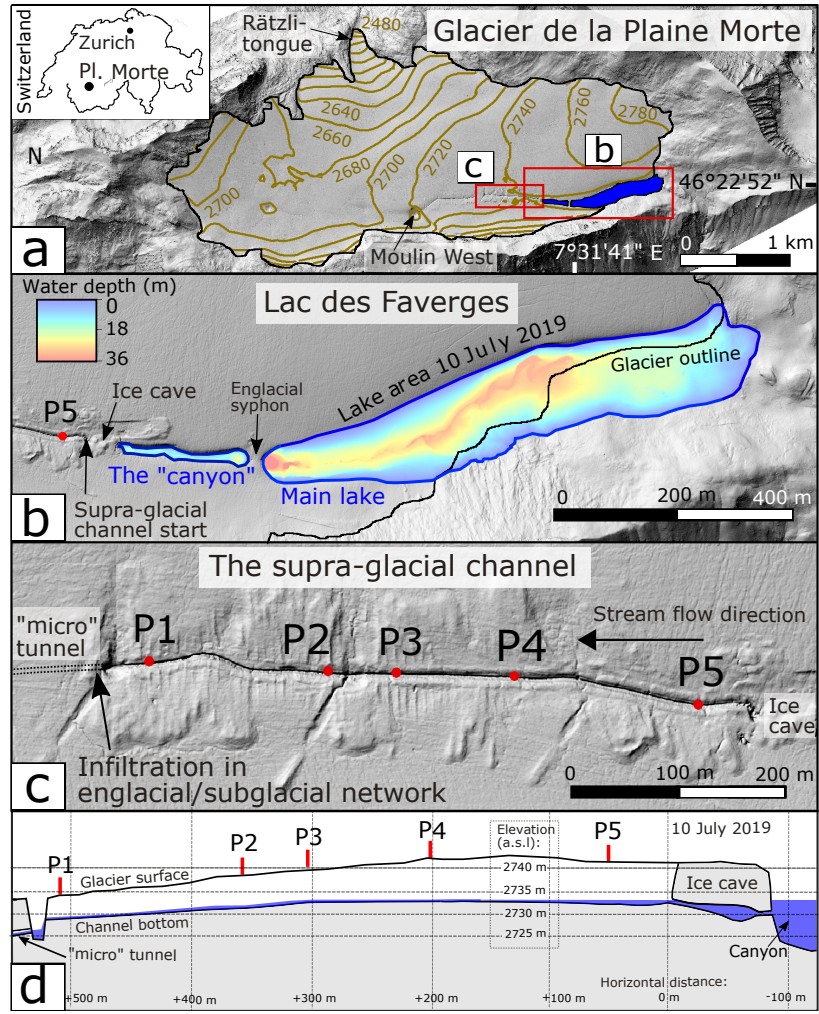

**Figure 1.** (a) Map of Glacier de la Plaine Morte where the ice-dammed Lac des Faverges lake is located. The lake area displayed in (b) corresponds to the maximum lake size reached on 10 July 2019 (at a lake level of 2733.15 m a.s.l). The supraglacial channel and the measurement stations P1 to P5 are presented in (c) and (d). The longitudinal profile in (d) was reconstructed from sparse elevation measurements of glacier surface and channel bottom prior to supraglacial lake drainage; the ice cave was not mapped and its representation is indicative only. The digital elevation model used in this figure was created from post-drainage aerial images acquired by the Swiss Federal Office of Topography on 3 September 2019 (see Section 4 for more information).

## 4 Methods

Two equations are of central importance to characterize the hydraulics and thermodynamics of the lake drainage through a supraglacial channel. The first is the Darcy-Weisbach equation (Incropera et al., 2007) which relates the water flow through a





channel to the gradient of the hydraulic potential and to the channel cross-sectional geometry:

$$\theta = \frac{f_D}{2g} \frac{v^2}{D_H},\tag{1}$$

where $g$ is the gravitational acceleration ($\mathrm{m\,s^{-2}}$), $\theta$ is the hydraulic slope (dimensionless, expressed as meter water-head drop per meter channel), $D_H$ the hydraulic diameter (m), and $v$ the stream velocity ($\mathrm{m\,s^{-1}}$). The constant of proportionality is the Darcy-Weisbach friction factor $f_D$.

The second equation characterises the thermodynamics, and relates the channel incision rate $m$ ($\mathrm{m\,s^{-1}}$) to heat flux $q$ ($\mathrm{W\,m^{-2}}$), where the latter can be related to the temperature difference between water and ice $\Delta T$ (°C) via the dimensionless Nusselt number $Nu$:

$$m = \frac{q}{\rho_i L_f} = \frac{1}{\rho_i L_f} \frac{k_w\,Nu\,\Delta T}{\lambda}.\tag{2}$$

Here, $k_w$ is the thermal conductivity of water ($\mathrm{W\,m^{-1}\,°C^{-1}}$), $\rho_i$ is the ice density ($\mathrm{kg\,m^{-3}}$), $L_f$ the latent heat of fusion

($\mathrm{J\,kg^{-1}}$) and $\lambda$ (m) is the length scale over which the turbulent heat transfer occurs (Incropera et al., 2007). Note that $Nu$ is not a constant but increases with discharge, and that the equations contain both physical constants (Table 1) and factors (Table 2) depending on the geometry ($\theta$, $D_H$), the hydraulics ($v$, $D_H$, $\theta$), or the thermodynamics ($T_w$). We will mirror this distinction in the description of the field measurements and the data processing.

In the following, we will describe our approach to obtain all terms of those two equations, in particular by determining their

dimensionless parameters $f_D$ and $Nu$, with our field measurements and their suitable processing steps.

### 4.1    Field measurements

#### 4.1.1    Topography

Topography of the lake and channel is necessary to characterize the geometry of the channel, to determine the watershed contribution to lake filling, and to obtain the lake bathymetry in order to relate volume changes to lake surface elevation

changes. To do so, we use Digital Elevation Models (DEMs) from the Swiss Federal Office of Topography (swisstopo). The latter were derived by using stereophotogrammetry, aerotriangulation, ground control points, and ADS100 Image strips (ground sampling distance of ∼0.1 m) acquired during flights commissioned by the Swiss Federal Office of the Environment on 28 August 2018 and 3 September 2019 when the lake was empty (see code and data availability section for references). The lake volume for the years 2012 to 2017 were also calculated using swisstopo DEMs. The theoretical nominal error of the swisstopo

DEMs is 2 m but is likely to be lower in the present situation with good ground contrast. These uncertainties on DEMs were not taken into account in the following.

#### 4.1.2    Water pressure, temperature and conductivity in the channel

Most measurements were conducted at five locations (called "stations") along the channel, named P1 to P5 (Fig. 1c and 1d). Stations P1 and P2 were marked with a stake drilled into the ice at the edge of the channel. At stations P3, P4 and P5 a cross-



beam was installed between stakes drilled on either side of the channel. Coordinates and elevation of the stations' stakes were measured by differential Global Positioning System (GPS) with an accuracy of $\pm 0.02$ m.

At four stations (P1,P2,P3 and P5), autonomous and time-synchronized data loggers (DCX-22-CTD by Keller) were installed to continuously recorded water pressure, temperature and conductivity. The logging interval was set to 1 s during tracer experiments and to 30 s otherwise. The accuracy of temperature measurements are $\pm 0.1°$C at stations P1 and P2, and $\pm 0.05°$C

at stations P3 and P5. We however note that this is a point measurement of the temperature with the sensor located at the floor of the channel, and that the bulk water temperature is therefore likely higher. The accuracy of the pressure measurements corresponds to a water column uncertainty of 0.2 m.

### 4.1.3 Channel geometry

At the stations, channel bottom elevation was measured using measuring tape either by lowering it from the top to the channel

bottom, or by abseiling with it into the channel. These measurements provide a longitudinal (i.e along the stream-flow direction) channel-elevation profile and were performed 11 times during the lake drainage. Estimated uncertainties are typically 0.1 m to 0.5 m in elevation and 1 m in horizontal position. The best accuracy in elevation was obtained for cross-beam stations (0.1 m). Channel width at the water flow surface was measured only infrequently at P3 and P5 due to complex accessibility, with an error of typically 0.1 m.

Higher temporal resolution of channel incision was obtained by measuring the clear diurnal melt-imprints left on the channel walls at P4 and P5 between 16 and 30 July 2019 (Fig. 4). We assume that the difference in elevation between two imprints represents the daily rate of channel floor erosion. This interpretation is supported by the observation that there were 14 marks over the 14-day observation period. We thus refer to these marks as daily water level cuts. Note that the channel geometry measurements described above give the temporal evolution of the channel, and thus the incision rates.

### 4.1.4 Hydraulics

To characterise the hydraulics of the channel we conducted measurements of discharge $Q$, the flow speed $v$, the stage $h$ (i.e water depth) in the channel, and the lake level $z_{\mathrm{lake}}$. In our notation, the variables used for an instantaneous measurement are marked with an index $i$ (e.g. $h_i$), to emphasize the difference with variables for continuous time series. Physical quantities for a spatial average between two stations are denoted with a bar (e.g. $\bar{w}$). Channel discharge $Q_i$ was measured using the salt

tracer dilution method (Hubbard and Glasser, 2005) at stations P1,P2 and P3. We carried out 33 salt injections at 12 different days during the campaign. Conductivity was measured at the monitored stations downstream of the salt injection location, with stations P3, P2, and P1 situated far enough downstream to ensure the required complete mixing of the tracer. Discharge can then be calculated from the conductivity measurements, as described in Section 4.2.2. The tracer experiments also provide information on travel time of the water between the stations equipped with conductivity sensors, and thus an average flow

speed $\bar{v}_i$ between stations can be calculated.





The water stage $h$ is measured via pressure measurements, which were corrected for atmospheric pressure variations. The measurements rely on the pressure transducer sinking to the bottom of the channel, which was the case for all presented measurements.

The elevation of the lake level $z_{\mathrm{lake}}$ was measured by two pressure transducers operated by Geopraevent, with a logging interval of 10 min. The position of the transducers was not always stable, probably due to icebergs shifting them; the resulting obvious shifts in the data were manually corrected, i.e. the data gaps were removed. The absolute elevation of the lake level was measured at three instances during the drainage using a differential GPS.

## 4.2 Data processing

### 4.2.1 Lake input from precipitation, snow and ice melt

Water input $Q_{\mathrm{in}}$ into the lake, by snow- and ice-melt or liquid precipitation, was substantial during the period of lake drainage but could not be directly monitored due to its non-localized nature. Instead, $Q_{\mathrm{in}}$ was estimated by using a distributed accumulation and temperature index melt-model driven by daily meteorological data (Hock, 1999; Huss et al., 2015). The model is calibrated with the seasonal mass balance measurements carried out by the programme Glacier Monitoring in Switzerland (GLAMOS). Seasonal mass balance is measured on Glacier de la Plaine Morte since 2009 using the direct glaciological method (GLAMOS, 2020). We applied the model with a daily resolution to the watershed of the lake, and used it to estimate $Q_{\mathrm{in}}$ consisting of snowmelt from the glacierized and ice-free portion of the basin, bare-ice melt and liquid precipitation. The location of the watershed over the gently-sloping glacier ice is inaccurately known, and was adjusted to match observed total lake volume on 10 July 2019 to the cumulative $Q_{\mathrm{in}}$ since the beginning of the melting season. The implicit assumption is, thus, that no water left the lake during that time span.

### 4.2.2 Hydraulics

The lake outflow $Q_{\mathrm{out}}$, which may consist of both supra- and subglacial runoff, was computed from lake level changes $\Delta z_{\mathrm{lake}}$ with a diurnal resolution ($\Delta t$) by considering (1) the lake surface area $A_{\mathrm{lake}}$ as a function of $z_{\mathrm{lake}}$ according to the DEM available for 28 August 2018, and (2) the recharge from melt $Q_{\mathrm{in}}$ at day $d$:

$$Q_{\mathrm{out},d} = Q_{\mathrm{in},d} - A_{\mathrm{lake},d} \frac{\Delta z_{\mathrm{lake},d}}{\Delta t}. \tag{3}$$

Instantaneous channel discharge $Q_i$ was determined at P3, P2 and P1 was from salt traces using the following steps. First, conductivity readings at those stations were converted into salt concentration using a calibration function derived from measurements conducted in laboratory. The function was derived by least-square regression of conductivity readings to salt concentration, for water temperature at 0°C and for concentration covering the entire range of observations. Second, salt concentrations were integrated over time for each injections and converted to discharge using the tracer dilution method (e.g. Hubbard and Glasser, 2005). We aimed to obtain a continuous discharge time series $Q$ by using the direct measurements $Q_i$ to calibrate a





stage-discharge relationship (or rating curve) at one station as follows

$$Q = a\,h^b, \tag{4}$$

where $a$ and $b$ are fitted parameters, and $h$ is the continuous time series of water stage at the selected station. The stage-discharge relation was established at P3 due to the high quality of direct discharge measurements by salt dilution, the most

continuous and reliable water stage time series, and the reasonably small geometry changes in the cross-section. Least-square fitting yielded parameters $a = 4.78 \pm 0.95$ and $b = 2.05 \pm 0.25$. The latter is in the range of literature values for natural rivers (Aydin et al., 2002, 2006). The resulting discharge was validated against 19 discharge measurements determined using salt dilution at different times and for stations not used in the calibration. This validation resulted in an root-mean-square-error of $0.11\,\mathrm{m^3\,s^{-1}}$, which is in line with the uncertainty in $Q$ estimated from the standard error of parameters $a$ and $b$.

A data gap in the channel's water stage time series between 13 and 24 July 2019 was filled with $Q$ based on daily lake discharge calculated according to Eq. (3). To do so, we make the hypothesis that the channel is the only drainage path existing for the lake water, i.e. that there is no subglacial drainage occurring during that time period.

The average hydraulic slope (or hydraulic gradient) $\bar{\theta}_i$ over a channel segment of length $l$ is

$$\bar{\theta}_i = \frac{\Delta p_i}{l}, \tag{5}$$

where $\Delta p_i$ is the difference of two hydraulic head measurements (i.e channel bottom elevation $z_i$ plus water stage $h_i$) at the beginning and at the end of the segment. We calculated $\bar{\theta}_i$, and subsequently derived quantities only for the segment P5-P3 because uncertainties in field measurements were the lowest for this part of the channel.

The hydraulic diameter $D_H$ is given by

$$D_H = \frac{4S}{P_w}, \tag{6}$$

where $S$ (m²) is the wetted cross-section and $P_w$ (m) is the wetted perimeter. To determine the Darcy-Weisbach friction factor $f_D$ (see Eq. 1), the hydraulic diameter over a channel segment at a given time, $\bar{D}_{H,i}$, needs to be determined. This is obtained by Eq. 6 using $\bar{S}_i$ and $\bar{P}_{w,i}$. $\bar{S}_i$ is given by dividing the discharge $\hat{Q}_i$ by the velocity $\bar{v}_i$, known at the times of salt dilution experiments. The channel width $\bar{w}$ is assumed to be constant between P3 and P5 as well as constant in time, and we found a value of $\bar{w} = 2 \pm 0.5$. In the following, we assume a rectangular cross-section, and define the mean wetted perimeter as

$\bar{P}_{w,i} = 2\bar{h}_i + \bar{w}$. This assumption is motivated by the initial channel shape (i.e. the shape prior to drainage) and by visual inspections that revealed a cross-sectional shape which did not evolve substantially over time.

The mean water stage $\bar{h}_i$ is calculated as $\bar{h}_i = \bar{S}_i/\bar{w}$. Alternatively, $\bar{h}_i$ can also be calculated as the mean of the water stage of two stations; both approaches lead to similar hydraulic diameters.

Finally, the friction factor $f_D$ is calculated from Eq. (1) with $\bar{S}_i$ and $\bar{D}_{H,i}$ between P5 and P3. Note that if we would consider

the channel cross-section to be a semi-circle, we would write $D_H = 2\sqrt{S_i/\pi}$, and $f_D$ would be on average 11% smaller. As an alternative to the Darcy-Weisbach friction factor, the Manning roughness law can be preferred to characterize the flow resistance (Clarke, 2003). The Manning roughness coefficient $n'$ ($\mathrm{m^{-1/3}\,s}$) can be calculated from $f_D$ by $f_D = 8gn'^2/R_H^{1/3}$, where the hydraulic radius $R_H$ is equal to $D_H/4$.





To quantify the turbulent flow, the Reynolds number $Re$ (dimensionless) is calculated at the single cross-section P3 using $\bar{w}$,
and both continuous discharge $Q$ and water stage $h$:

$$Re = \frac{vD_H}{\nu}. \tag{7}$$

Here, $v = Q/S$, with $S = h\bar{w}$, whilst $D_H$ is calculated at P3 using Eq. (6) and $\nu$ is the kinematic viscosity ($\mathrm{m^2\,s^{-1}}$).

### 4.2.3 Thermodynamics

The Nusselt number $Nu$, i.e. the unknown parameter in Eq. (2), is defined as the ratio between convective and conductive heat
transfer across the water-ice interface:

$$Nu = \frac{h_t \lambda}{k_w}, \tag{8}$$

where $\lambda$ (m) is the length scale over which the convective heat transfer occurs, $h_t$ is the convective heat transfer coefficient
($\mathrm{W\,m^{-2}\,{}^\circ C^{-1}}$), and $k_w$ is the thermal conductivity of water ($\mathrm{W\,m^{-1}\,{}^\circ C^{-1}}$). For $\lambda$ we use the typical hydraulic diameter of the
channel $D_H$, which is often used in glaciology (Clarke, 2003; Sommers and Rajaram, 2020) and other fields (Incropera et al.,
2007; Shah and London, 1978). Note that this choice of $\lambda$ is different to $P_w$, which was used by Walder and Costa (1996) when
simulating ice-dam breaches.

Since the hydraulic diameter strongly depends on the channel width in the case of a broad channel, it is relatively poorly
constrained in our study. We therefore define $\lambda$ as the typical, and constant, hydraulic diameter which we calculate using Eq. (6).
With $\bar{h}$ the typical water stage observed in the channel ($\bar{h} = 0.5 \pm 0.1\,\mathrm{m}$) we obtain $P_w = 3 \pm 0.54\,\mathrm{m}$ and $\bar{D}_H = 1.30 \pm 0.22\,\mathrm{m}$.
For comparison, $\bar{D}_{H,i}$ calculated above ranges between $1.0\,\mathrm{m}$ to $1.6\,\mathrm{m}$, with a mean value of $1.26\,\mathrm{m}$. We then use $\lambda = \bar{D}_H$ to
obtain $Nu$ in Eq. (8).

In glaciological applications and elsewhere, $Nu$ is usually calculated using an empirical relation, often the Dittus-Boelter
equation (e.g. Clarke, 2003; Spring and Hutter, 1981; Nye, 1976), or the Gnielinski correlation (e.g. Ancey et al., 2019). These
two equations parameterise $Nu$ using the Reynolds ($Re$) and Prandtl ($Pr$) numbers, where the latter is the ratio of the dynamic
viscosity to the thermal diffusivity of water ($Pr = 13.5$ at $0^\circ$C, Clarke (2003)). The Dittus-Boelter equation reads

$$Nu = A\,Pr^\alpha Re^\beta, \tag{9}$$

where $A$, $\alpha$ and $\beta$ are empirical coefficients given in the literature (Incropera et al., 2007). The Gnielinski correlation addition-
ally uses $f_D$ and reads

$$Nu = \frac{\frac{f_D}{8}(Re - 1000)Pr}{1 + 12.7(\frac{f_D}{8})^{\frac{1}{2}}(Pr^{\frac{2}{3}} - 1)}. \tag{10}$$

The available measurements allow us to calculate $Nu$ using two different methods (see below), and thus we can compare
our findings to the above empirical equations. The first method, termed the *melt-rate method*, considers the melt rate and water
temperature at one location as a function of time. $Nu$ is then directly derived from Eq. (2) with the vertical melt rate $m$ given





by repeated channel floor elevation measurements and with the water temperature given by the continuous monitoring. The water temperature measurements are averaged over the time span between two channel elevation measurements, therefore, $Nu$
obtained by using the melt-rate method is time-averaged as well.

The second method, termed the *spatial-cooling rate method*, considers the water temperature at an instance in time and its decrease as a function of distance along the channel. The water temperature $T_w(x)$ decreases following an exponential law (e.g. Isenko et al., 2005), which can be derived from energy conservation (see Appendix A) and can be written as

$$T_w(x) = T_0 \, e^{-\frac{x}{x_0}}, \tag{11}$$

where $x$ is the distance (m) from the origin (in our case P5, the uppermost monitoring station), $T_0$ is the temperature at this location (°C), and $x_0$ is the e-folding length (m). Physically, $x_0$ is the distance over which the temperature decreases by a factor e and can be expressed in terms of $Nu$, $Q$, and the wetted perimeter $\bar{P}_w$ as

$$x_0 = \frac{Q \, c_w \rho_w \lambda}{Nu \, k_w \, \bar{P}_w}, \tag{12}$$

where $c_w$ is the specific heat capacity of water ($\mathrm{J\,°C^{-1}\,kg^{-1}}$). We obtain $x_0$ from a least-square fit of Eq. (11) to the hourly-
averaged temperature at stations P5, P3, P2 and P1 and, thus, $Nu$ can be calculated.

### 4.2.4 Uncertainty propagation

In this study, uncertainties of field measurements come from the sensors' sensitivity and limitations of the measurement procedures. Both uncertainties are quantified and propagated through the equations by using a Monte-Carlo approach (Carlson, 2020). Since this allows us to propagate errors faithfully also through non-linear functions, results are systematically presented
with their standard deviation.

**Table 1.** Physical constants used in this work. If not specified, constant refers to the property of water at at 0°C.

| Physical constants | Var. | Value | Units |
|---|---|---|---|
| Density of ice | $\rho_i$ | 900 | $\mathrm{kg\,m^{-3}}$ |
| Latent heat of fusion | $L_f$ | $333{\times}10^3$ | $\mathrm{J\,kg^{-1}}$ |
| Density | $\rho_w$ | 1000 | $\mathrm{kg\,m^{-3}}$ |
| Specific heat capacity | $C_w$ | $4.18{\times}10^3$ | $\mathrm{J\,°C^{-1}\,kg^{-1}}$ |
| Thermal conductivity | $k_w$ | 0.57 | $\mathrm{W\,m^{-1}\,°C^{-1}}$ |
| Kinematic viscosity | $\nu$ | $1.8{\times}10^{-6}$ | $\mathrm{m^2\,s^{-1}}$ |
| Prandtl number | $Pr$ | 13.5 | - |



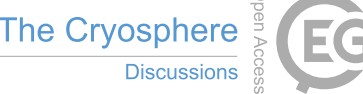

**Table 2.** Table of variable names. *salt dil.* stands for "salt dilution experiment technique", and *a.s.l* stands for "above sea level". Individual quantities might either be available for a point in time ($PT$; the index $i$ means that the measurement is instantaneous) or as a time series ($TS$), or might be constant through time ($CT$). A bar over the corresponding symbol indicates that the quantity is averaged over a given channel segment, i.e. between two stations.

| Direct field Measurements | Notation | | | Unit |
|---|---|---|---|---|
| | PT | CT | TS | |
| *of channel:* | | | | |
| Water stage | $h_i$ | | $h$ | m |
| Channel floor elevation | $z_i$ | | | m (a.s.l) |
| Hydraulic head | $p_i$ | | | m |
| Hydraulic slope | $\theta_i$ | | | - |
| Width | $w_i$ | | | m |
| Discharge (salt dil.) | $Q_i$ | | | $\mathrm{m^3\,s^{-1}}$ |
| Stream velocity (salt dil.) | $\bar{v}_i$ | | | $\mathrm{m\,s^{-1}}$ |
| Wetted cross-section (salt dil.) | $\bar{S}_i$ | | | $\mathrm{m^2}$ |
| Water temperature | | | $T_w$ | °C |
| *of lake:* | | | | |
| Lake level | | | $z_\mathrm{lake}$ | m |
| **Derived & other variables** | | | | |
| *of channel:* | | | | |
| Melt rate | $m$ | | | $\mathrm{m\,s^{-1}}$ |
| Discharge | | | $Q$ | $\mathrm{m^3\,s^{-1}}$ |
| Stream velocity | | | $v$ | $\mathrm{m\,s^{-1}}$ |
| Wetted cross-section | | | $S$ | $\mathrm{m^2}$ |
| Wetted perimeter | $P_{w,i}$ | | $P_w$ | m |
| Hydraulic diameter | $D_{H,i}$ | | $D_H$ | m |
| Reynolds number | | | $Re$ | - |
| Darcy Weisbach friction factor | $f_D$ | | | - |
| Manning roughness | $n'$ | | | $\mathrm{m^{-1/3}\,s}$ |
| Convective heat transfer | $h_t$ | | | $\mathrm{W\,m^{-2}\,°C^{-1}}$ |
| Heat flux | | | $q$ | $\mathrm{W\,m^{-2}}$ |
| Energy content of water | | | $E$ | $\mathrm{J\,m^{-1}}$ |
| Energy source term | | | $M$ | $\mathrm{W\,m^{-1}}$ |
| e-folding length of water temperature decrease | | | $x_0$ | m |
| Nusselt number | | | $Nu$ | - |
| *of lake:* | | | | |
| Lake surface area | | | $A_\mathrm{lake}$ | $\mathrm{m^2}$ |
| Lake inflow | | | $Q_\mathrm{in}$ | $\mathrm{m^3\,s^{-1}}$ |
| Lake outflow | | | $Q_\mathrm{out}$ | $\mathrm{m^3\,s^{-1}}$ |
| **Time independent variable** | | | | |
| Length between two stations | | $l$ | | m |
| Distance in channel from P5 | | $x$ | | m |
| **Time and space independent parameter** | | | | |
| *of channel:* | | | | |
| Mean width | | $\bar{w}$ | | m |
| Mean water stage | | $\bar{h}$ | | m |
| Mean water perimeter | | $\bar{P}_w$ | | m |
| Mean hydraulic diameter | | $\bar{D}_H$ | | m |
| Nusselt length scale | | $\lambda = D_H$ | | m |





## 5   Results

### 5.1   Lake drainage hydrographs 2012-2019

Figure 2 shows the temporal evolution of lake-water volumes between 2012 and 2019, and hourly-averaged discharge. The lake water input $Q_{\mathrm{in}}$ was only accounted for the year 2019, since it becomes relevant to take it into account in the calculation

of $Q_{\mathrm{out}}$, due to much smaller value of the latter than previous year. The increasing trend of both maximum volume and peak discharge from 2012 to 2018 is clearly visible. The drainage onset time depends, amongst others, on the meteorological conditions during the lake filling phase. With high air temperatures, the date of complete filling of the lake basin occurs earlier. Also, an early depletion of the winter snow coverage is likely to be linked with an early development of the subglacial drainage system, which in turn favours subglacial lake drainage (GLAMOS, 2018). Since 2014, the lake volume has systematically

reached more than $2 \times 10^6 \, \mathrm{m}^3$, and the subglacial release of the total lake water occurred within a few days, except for 2015 when the water drained through a supraglacial channel into a nearby moulin for about two weeks.

The 2019 lake drainage pattern is drastically different from previous years due to the artificial intervention. It can be characterized by four distinct phases. The first phase (Phase I) is from 10 July to 1 August 2019, when approximately half of the lake emptied through the supraglacial channel. Phase II is from 2 to 15 August 2019, when the lake level remains roughly constant

but lake water was still running in the channel. Phase III is from 15 to 21 August 2019, when lake water stopped running in the channel and the lake level remained constant or slightly increased. Phase IV is from 22 to 27 August 2019, when the second half of the lake volume emptied subglacially, similar to the natural drainage mechanism of previous years. Note that the discharge peak of $3.5 \, \mathrm{m}^3 \mathrm{s}^{-1}$ was much lower compared to other years, e.g. over 20 times lower than in 2018 (Fig. 2). In this regard, the technical intervention was very successful.

### 315   5.2   Channel geometry

The vertical channel incision at five locations along the channel is presented in Fig. 3. This incision shows a uniform spatial pattern, and is about 8 m during the supraglacial drainage (phase I and II: 10 July - 15 August 2019). Note that the channel slope was not uniform prior to the drainage onset (Fig. 1d), and that it remained relatively constant after natural adjustments during the first days of drainage. The low slope at the channel segment P5–P3 leads to uniform stream flow, that allowed the formation

of clear daily water level cuts (Fig. 4). In contrast, the more turbulent water flow between P2 and P1 (higher slope) led to the formation of step-pools (e.g. Vatne and Irvine-Fynn, 2016). Note that no meandering (Karlstrom et al., 2013) occurred and, thus, the channel length stayed constant.

Widening is substantial only at P5 where channel width increased from 1 m on 10 July to $3 \pm 0.1 \, \mathrm{m}$ on 8 August 2019. Further downstream, e.g. at P4, the widening is minor (Fig. 4). Overall, the channel geometry was mainly driven by vertical

incision rather than lateral melting.

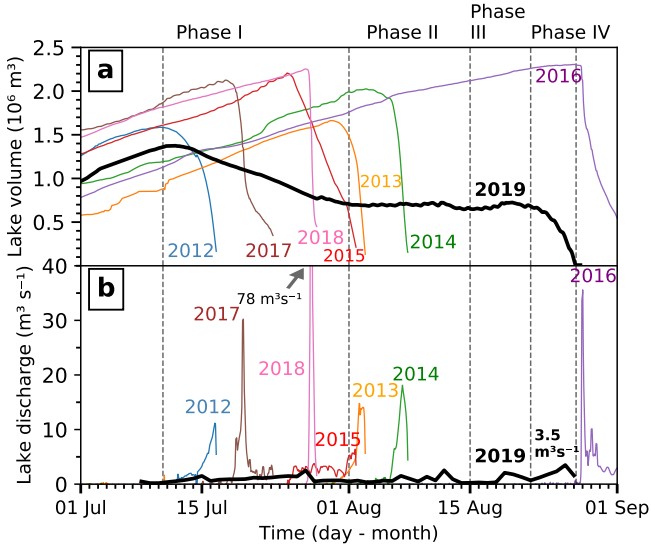

**Figure 2.** Lake volume (a) and lake discharge (b) during summer from 2012 to 2019. The year 2019 is represented by a black and thicker line. Discharge is shown as hourly averages for 2012 to 2018, and daily averages for 2019. Note that the 2018 discharge peak (78 $m^3 s^{-1}$) is beyond the plotted range.

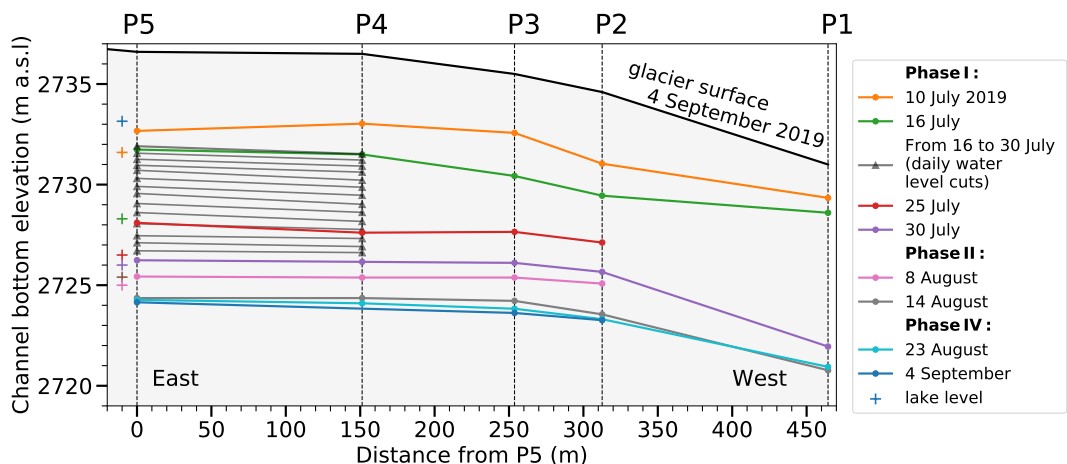

**Figure 3.** Channel bottom elevation at P5, P4, P3, P2 and P1 during the lake drainage (locations are presented in Fig. 1). The color-coded crosses indicate lake level at the corresponding time. Note that the actual lake was located further east than P5. Distance between stations are taken along the channel flow path.



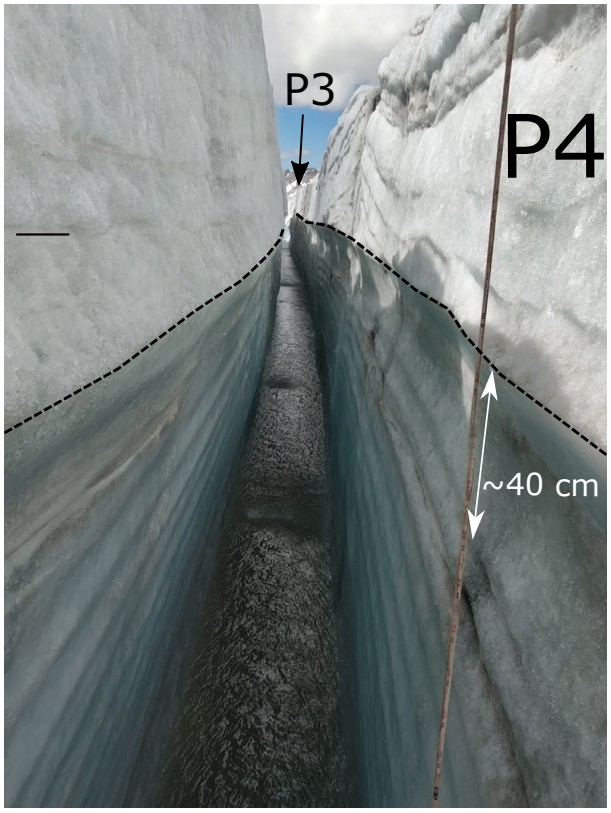

**Figure 4.** Photo from within the supraglacial channel from P4 towards P3. The dashed line represents the highest water stage in the channel (10 July 2019) prior to the supraglacial lake drainage start and prior to the onset of channel-bottom incision. Daily water level cuts are clearly visible on both sides. The picture was taken on 30 July 2019.

## 5.3   Channel hydraulics

Water stage (Fig. 5) and the hydraulic gradient determine the water discharge. Water stage measurements were challenging during the first days of the supra-glacial drainage (i.e. 10 July 2019, beginning of Phase I) since the excavator used to deepen the channel spillway left irregular traces on the channel bottom. Nevertheless, probe measurements (at P5, P4, P3, P2 and P1) together with water pressure sensors (at P5, P3, P2 and P1 only) reveal that water stage on 10 July 2019 was around 1 m at P5, 0.4 m at P4 (which was close to the spillway location), and 0.3-0.5 m for the others stations. Water stage stabilized at 0.6 m after a few days of drainage (Phase I) at P5, and at around 0.4-0.5 m at the others stations. Daily fluctuation were typically of 0.1 m due to the daily melting cycle influencing the lake input. At P1, measurements were soon no longer feasible because of the formation of step pools. Water stage slowly decreased to a value of 0.1-0.2 m uniformly over the channel during Phase III and Phase IV, when lake water no longer drained through the channel.



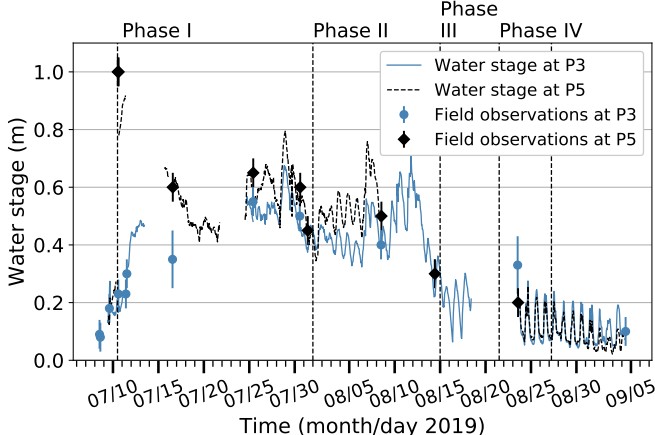

**Figure 5.** Hourly time series of water stage at P5 (dashed black line) and P3 (blue line) from continuous pressure measurements. Direct observations (and associated standard errors) from field visits are marked by black diamonds and blue dots, respectively. Part of the discrepancies between direct field observations and continuous water stage from pressure sensors can be explained by the probing not always being made at the exact location of the sensors.

Figure 6 presents time series of water temperature, channel discharge and channel-bottom elevation at station P3, along with the lake level. Data gaps in discharge and temperature times series are due to disruptions of logging. The daily mean discharge time series between 13 and 24 July 2019 has been calculated using lake level change and modelled lake inflow (see Section 4). The peak in channel discharge was reached on 14 July 2019 (Phase I), with a daily average of $1.7\,\mathrm{m^3s^{-1}}$. The good agreement
between the different direct discharge measurements ($\hat{Q}_i$) and the continuous discharge at P3 indicate that the stage/discharge relationship is valid over the entire drainage duration. The decoupling between $Q$ at P3 and $Q_\mathrm{out}$ during Phase III and IV (Fig. 6b) is because the lake did no longer drain through the channel at that stage. The temperature and discharge signal of the lake water is clearly visible in the channel until 14 August 2019. This is in contrast with the temperature signal from the glacier's daily melt pattern from 23 August to 4 September 2019 (water temperature close to 0°C at night), when the lake was
no longer draining through the channel (beginning of Phase III).

The stable mode of drainage during Phase I is corroborated by the observation that the distance between daily water level cuts in the channel (indicative for channel floor erosion, cf. Sec. 4.1.3) corresponds to the rate at which the lake level lowers (Fig. 6). At P5, this distance varies between 30 cm and 60 cm per day during the second half of July 2019, and drops on average to 12 cm per day for the first half of August 2019.
The stream flow in the channel is highly turbulent during supraglacial drainage. The Reynolds number fluctuates with discharge, and is between $2.5\times10^5$ and $1.5\times10^6$ (Phase I and II). Note that the transition between laminar flow in the lake and turbulent flow somewhere at the channel entrance is further upstream than P5 (the location is not known exactly).

The Darcy-Weisbach friction factor was calculated for the channel segment between P5 and P3, where accurate calculation of $\bar{D}_H$ and thus $\bar{P}_w$, was possible. The time series of the inferred friction factor is presented in Table 3.

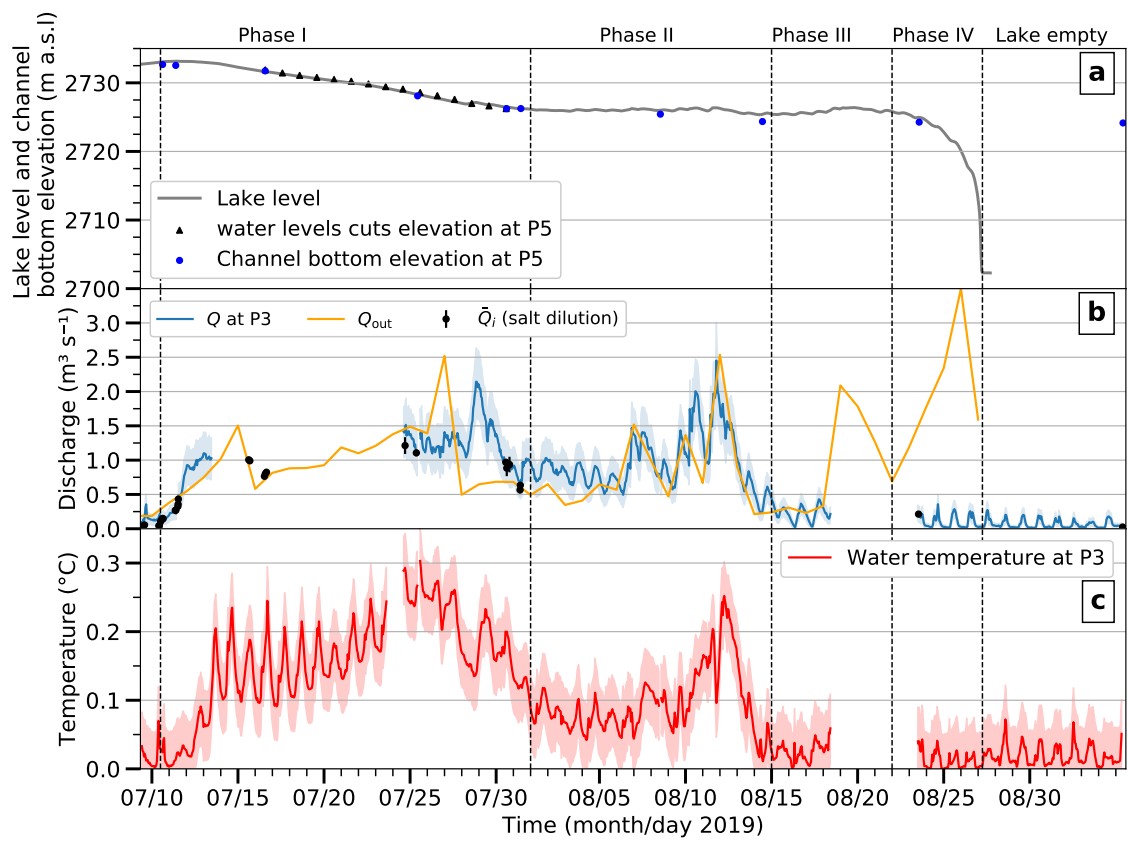

**Figure 6.** (a) Lake level elevation (continuous curve), channel bottom elevation (blue dots) and water level cut elevations (black dots) at P5. (b) Hourly channel discharge at P3 ($Q$), direct discharge measurement by salt dilution averaged over all stations ($\bar{Q}_i$), and lake discharge from elevation change ($Q_{\text{out}}$). (c) Water temperature at P3. The four distinct lake drainage phases are delimited by the dashed line and explained in the main text (Sec. 5.1). Standard uncertainties for hourly discharge and temperature are shown with light bands.

## 5.4 Thermodynamics

Water temperature, measured continuously at P5, P3, P2, and P1, shows an exponential decrease along the channel (Fig. 7), and exhibits daily fluctuations (Fig. 6, e.g. $\pm 0.1$°C during Phase I). The relation given in Eq. (11) is fitted to these temperature observations. Note that the pattern is similar whenever lake water was flowing through the channel.



**Table 3.** Darcy-Weisbach friction factor $f_D$ and Manning roughness $n'$ with associated standard deviation in the channel segment between stations P5 and P3. All measurements were performed during Phase I of the drainage when salt dilution experiments were conducted (see Fig. 6).

| Date and time (2019) | $f_D$ (-) | $n'$ (m$^{-1/3}$ s) |
|---|---|---|
| 11 July, 09:27 | $0.41 \pm 0.12$ | $0.055 \pm 0.009$ |
| 11 July, 12:51 | $0.34 \pm 0.10$ | $0.051 \pm 0.008$ |
| 16 July, 14:12 | $0.17 \pm 0.02$ | $0.038 \pm 0.003$ |
| 25 July, 08:44 | $0.17 \pm 0.02$ | $0.040 \pm 0.003$ |
| 30 July, 14:47 | $0.48 \pm 0.07$ | $0.068 \pm 0.005$ |
| 30 July, 18:34 | $0.19 \pm 0.03$ | $0.042 \pm 0.003$ |
| 31 July, 09:39 | $0.35 \pm 0.06$ | $0.056 \pm 0.005$ |
| Mean | $0.30 \pm 0.12$ | $0.050 \pm 0.011$ |

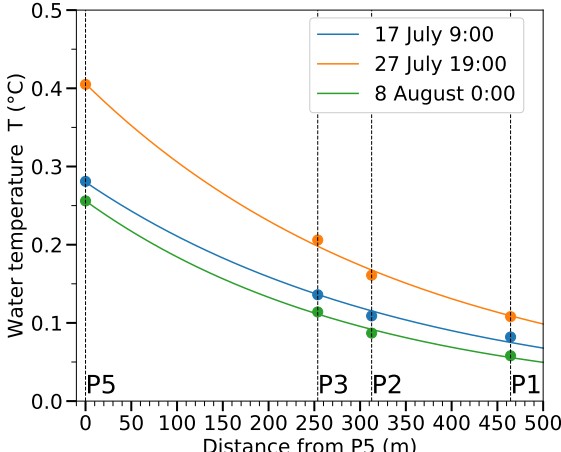

**Figure 7.** Temperature decrease along the channel-flow path at three different times. Dots are observations, lines correspond to the fitted exponential law according to Eq. (11).

Figure 8 presents time series of the Nusselt number *Nu* calculated from our measurements according to the melt-rate method (Eq. (2)), the spatial-cooling rate method (Eq. (12)), as well as from the empirical Dittus-Boelter equation (Eq. (9)) and the Gnielinski correlation (Eq. (10)). For the Dittus-Boelter equation we use coefficients $A$, $\alpha$ and $\beta$ from previous studies (Table 4); for the Gnielinski correlation, we use the mean friction factor $f_D = 0.3$ (Table 3). The heat transfer is dominated by convection, with typical *Nu* values in the order of $10^4$. The results from the melt-rate and spatial-cooling rate methods are in good agreement, except for the period 11-16 July 2019 (beginning of Phase I). For this period, the higher *Nu* values obtained





by the melt rate method compared to the spatial-cooling rate method could be explained by the very high melt rate at P3 (see Fig 3).

Our values for *Nu* are significantly higher than the ones derived from the Dittus-Boelter equation using standard coefficients (e.g. Clarke, 2003). We note, however, that the coefficients used by Lunardini et al. (1986) and Vincent et al. (2010) result in *Nu*-values that lie at the lower and upper edge, respectively, of the uncertainty range of our *Nu*-values. Conversely, the

Gnielinski correlation produces values of *Nu* that are significantly higher than our results, as well as the ones obtained by the alternative methods.

**Table 4.** Dimensionless coefficients of the Dittus-Boelter equation (Eq. (9)) from various studies, including this one (see Section 4.2.3).

| Study | $A$ | $\alpha$ | $\beta$ |
|---|---|---|---|
| Standard (e.g. Clarke, 2003) | 0.023 | 2/5 | 0.8 |
| Lunardini et al. (1986) | 0.0078 | 1/3 | 0.927 |
| Vincent et al. (2010b) | 0.332 | 1/3 | 0.74 |
| This study | 1.78 | 1/3 | 0.58 |

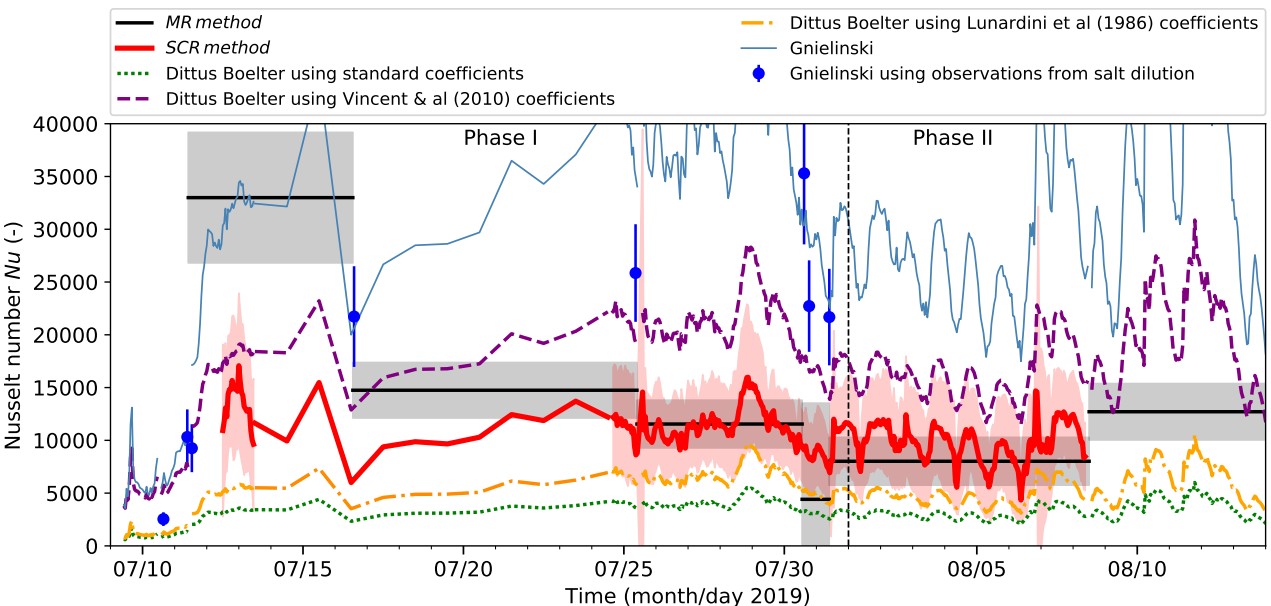

**Figure 8.** Nusselt number *Nu* at P3 computed using two different approaches and several empirical relations. *MR method* refers to melt-rate method (Eq. (2)) and *SCR method* refers to spatial-cooling rate method (Eq. (12)),. The Dittus Boelter equation and the Gnielinski correlation are presented in Eq. (9) and Eq. (10), respectively. For the sake of readability, the standard deviation (band around the mean values) is only displayed for the values derived from our measurements. The vertical dashed line separates Phase I and II of the lake drainage.





The Nusselt number $Nu$ is dependent on the Reynolds number $Re$ via turbulent mixing. Figure 9a shows the relation between $Nu$ and $Re$ as determined by our field measurements, along with the previously used parameterisations for $Nu$. Of note is that our results show a less pronounced dependence of $Nu$ on $Re$ than other assessments. Indeed, fitting the Dittus-Boelter equation

(coefficients $A$ and $\beta$, Eq. (9)) to our data yields an exponent of $\beta = 0.58$, which is low compared to exponents between 0.75 and 0.93 found by Clarke (2003), Lunardini et al. (1986) and Vincent et al. (2010b) (Table 4).

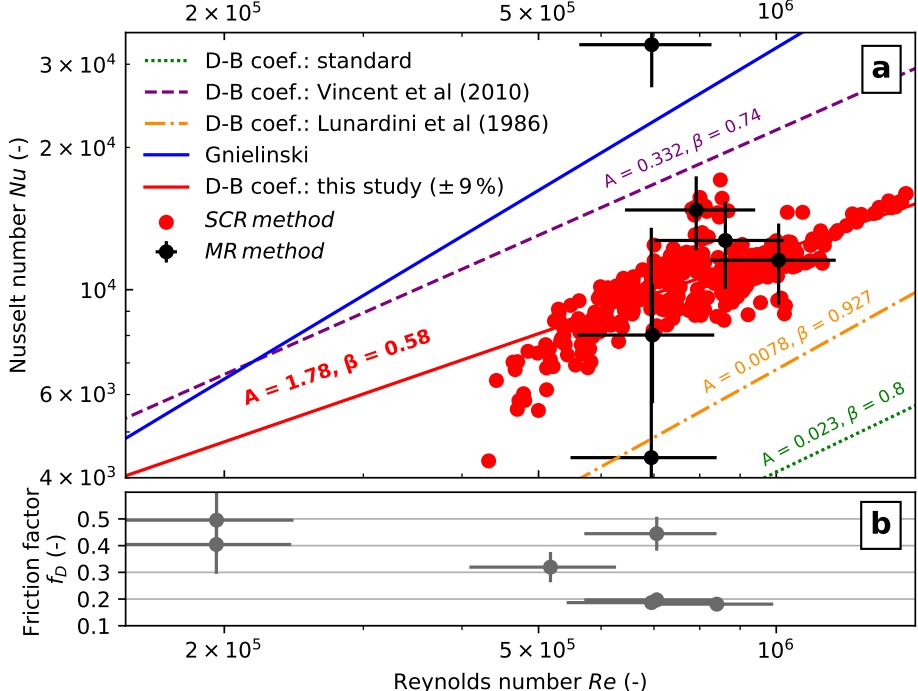

**Figure 9.** Nusselt number (a) and friction factor (b) against Reynolds number (Eq. (7)). Note that all quantities are dimensionless. The Nusselt number from our observations is calculated using the *melt-rate method* at P3 (Eq. (2)) and the *spatial-cooling rate method* (Eq. (12)). The band shows the mean relative error of 9%. The Nusselt number is also calculated from the Dittus-Boelter equation Eq. (9)) using coefficients (named "D-B coef." in the legend) from other studies (see Table 4) and the Gnielinski correlation (Eq. (10)).

## 6 Discussion

We provided an extensive data set of a supraglacial lake drainage through a channel, and characterise its hydraulics and thermodynamics. We derive key parameters, namely the factors of hydraulic friction and heat transfer. In the following, we

discuss the implications of our findings to future studies and for hazard mitigation measures.





## 6.1 Reconstruction of the lake drainage during Phase I–IV

The four phases of the lake drainage are interpreted as follows: Phase I is characterized by stable supraglacial lake drainage, i.e. the lake draw-down is controlled by the rate of vertical channel incision. There is a significant difference between the computed lake outflow $Q_{\mathrm{out}}$ and the channel discharge $Q$ at P3 during the sub-period 26 July to 30 July 2019 (Fig.6b) coinciding with

strong rain that ended a heat wave which started on 20 July 2019.

During Phase II, the lake level remained constant (Fig.6a), but the lake water was still running in the channel as evidenced by the relatively warm water (Fig.6c). This is indicative for the outflow of the lake-channel system behaving like a non-erosive spillway: the channel only received the water from snow and ice melt, which spilled above a constant elevation. The exact location of the spillway is unknown: it could either be located in the englacial siphon between the main lake and the canyon,

or in the ice cave (Fig. 1) although visual inspection gave no evidence for the latter. Phase III is characterized by the stop of the supraglacial drainage around 15 August 2019. It is likely that lake surface became too small, such that the lake draw-down became higher than the channel incision. This is especially likely if the channel was disconnected from the main lake by the spillway between the canyon and the main lake, or in the case of subglacial leaks. Although no sensors were installed in the channel from 19 to 23 August 2019, the channel incision between 14 to 23 August 2019 was negligible (Fig. 3). This indicates

that the peak in $Q_{\mathrm{out}}$ on 19 August 2019 was due to subglacial rather than supraglacial drainage.

During Phase IV, the lake emptied subglacially, with no influence from the supraglacial channel. The triggering mechanism is presumably similar to previous years. Lindner et al. (2020) showed, for example, that the drainage in 2016 was initiated by hydrofracturing.

## 6.2 Hydraulics

We calculated the hydraulic friction factor $f_D$ of the channel at several instances during Phase I (Table 3), when the necessary salt-dilution measurements are available. Values for $f_D$ range from 0.17 to 0.48, corresponding to a Manning roughness $n'$ of 0.038 to 0.068 m$^{-1/3}$ s. These values are similar to the ones of Mernild et al. (2006), who inferred $n'$ between 0.036 and 0.058 m$^{-1/3}$ s in supraglacial streams on the Greenland Ice Sheet. Similar observations of Gleason et al. (2016) revealed strong variability of $n'$ in time and space, ranging from 0.009 to 0.154 m$^{-1/3}$ s with a mean value of 0.035$\pm$0.027 m$^{-1/3}$ s, similar to

ours. The lower end of their range corresponds to a smooth channel, whereas the upper end cannot be explained by ice-channel roughness alone and is indicative for, e.g., suspended slush-ice being present in the channel.

Why the friction factors vary so much is unclear from our observations. For instance, there is no correlation with the Reynolds number (Fig. 9b). Previous studies already highlighted the need to better quantify hydraulics of englacial channels (e.g. Clarke, 2003; Gleason et al., 2016), and the need for additional in situ observations to better constrain the parameters that control

discharge (e.g. Kingslake et al., 2015; Smith et al., 2015). Our work provides a range of accurate, field-based friction factors for a supraglacial stream at different times during a lake drainage, and shows its variability. However, the large range of friction factor values reported here and in other studies suggests that using a constant value in modelling studies could be inappropriate.





Instead, we suggest that modelling studies should treat $f_D$ as a stochastic variable (e.g. Brinkerhoff et al., 2021; Irarrazaval et al., 2021), i.e. that they should use a range of values as opposed to a single one.

### 6.3 Thermodynamics

In general, the supraglacial drainage of an ice-dammed lake progresses via the incision of the channel by melt. The incision rate determines whether the lake drains gradually, i.e. with an approximately constant discharge over time, or unstably, i.e. with a progressively increasing discharge (Raymond and Nolan, 2000).

To characterise channel incision, the heat transfer between the advected lake water and the ice channel walls needs to be quantified. This heat transfer is captured by the Nusselt number $Nu$, which was derived from measurements in this study. Our results (Fig. 8 and 9) are in-between the predictions of the Dittus-Boelter equation (Eq. (9)) using the parameters from Vincent et al. (2010) which are higher than our values and Lunardini et al. (1986) with lower values than our results. This suggests that using the parameterisations of those two studies, that have a cryospheric context as well, gives an interval of Nusselt numbers to consider. Again, as for the hydraulic friction factor, the large range of plausible Nusselt numbers means that it should be a stochastic parameter in modelling studies.

The cause of the discrepancy between the exponent of Reynolds number in the Dittus-Boelter equation determined in our study ($\beta = 0.58$) and others ($\beta$ between 0.75 and 0.93, Table 4, Fig. 9a) is not clear. Our water temperature measurements were conducted using CTD sensors which sunk to the channel bottom. Their close proximity to the ice means that they might have measured a temperature below the bulk water temperature, which would lead to an overestimation of $Nu$ using the melt-rate method. Conversely, the results of the spatial-cooling rate method would likely be less impacted as the temperature e-folding length should not depend on the location of the sensors. Future studies should put an emphasis on accurate water temperature measurements, for instance by using fiber optics methods as in Karlstrom et al. (2014), paying attention to accurately estimate the bulk water temperature.

Longitudinal temperature profiles of supraglacial channels have been studied in both the field and in the laboratory, but only by one study so far (Isenko et al., 2005). Water temperature decreases exponentially with distance, which is the basis of estimating $Nu$ though our spatial-cooling rate method (Eq. (12)). This method is useful because it is easier to implement and allows a higher sampling rate than the melt-rate method (Eq. (2)) because it only requires temperature measurements and not the more challenging channel-incision rate measurements, as were used in other studies (e.g. Vincent et al., 2010). If the melt-rate method is chosen, direct measurements of channel incision are needed and we support the idea of Raymond and Nolan (2000) that daily water level cuts can be used to conduct those measurements a posteriori (Fig. 4).

Besides the incision rate, the channel aspect-ratio plays a major role in the drainage stability of a supraglacial lake: for a given stage, a wider channel has a bigger discharge capacity than a narrower channel, and will consequently contribute to stabilize the drainage by accelerating the lake-level draw-down (Raymond and Nolan (2000), their Eq. (8)). Channel width was considered constant in the present field study and was used to calculate the hydraulic diameter and consequently all parameters which are derived from it. The large uncertainties we put on the width take into account a potential change in width, and the



estimated parameters take this fully into account. Moreover, our stage/discharge relation at P3 seems to work well despite of a slight widening at that location.

The channel aspect-ratio was considered in the model of Jarosch and Gudmundsson (2012) which suggests that this ratio is determined by melt rate dependence on water depth: if the melt rate is independent of water depth, a very broad channel forms,

whereas if it scales linearly, a nearly semi-circular channel forms. To our knowledge, there is no theoretical work available for how melt rates distribute over the channel perimeter, but an extension of Sommers and Rajaram (2020) could potentially be applied to this issue. Future field-based studies could try to quantify the channel aspect-ratio, as it would give indications on both lake drainage stability as well as melt-rate distribution along the channel perimeter.

### 6.4  Hazard mitigation

The maximum lake volume reached in 2019 was limited by the constructed supraglacial channel to about two thirds of its potential volume, and half of the lake water ($\sim$0.7x10$^6$ m$^3$) drained, stably, through the channel once the lake overspilled into it. In this sense, the hazard mitigation of Lac des Faverges was very successful; however construction costs were considerable at 1.7 million CHF. Furthermore, construction costs are relatively low compared to the damage costs of 2.5 million CHF only in 2018. The relatively small volume of remaining water later then drained subglacially during an outburst event (Fig. 2). The

peak discharge was only $\sim$3.5 m$^3$s$^{-1}$, and thus far lower than the $\sim$80 m$^3$s$^{-1}$ recorded in 2018.

The intervention at Lac des Faverges thus indicates that a channel dug at the glacier surface prior to the lake filling can successfully limit the maximum lake volume and thus the hazard potential. Indeed, partially or fully-filled lakes have been successfully drained in the past via such channels (e.g. Vincent et al., 2010), however not at the scale of the present artificial intervention. Nonetheless, surface lake drainages also have considerable hazard potential (e.g. Ancey et al., 2019; Walder

and Costa, 1996) and therefore the prediction of whether a surface drainage proceeds stably or unstably is critical for hazard assessments. Raymond and Nolan (2000) proposed a criterion based on lake area and temperature (their Eq. (8)). The area and temperature of Lac des Faverges respected this stability criterion when the drainage initiated in 2019, but only barely. However, the past surface drainage of 2015 (Fig. 2), which drained stably through a shorter channel (about 400 m), suggests that the 2019 drainage was well in the stable regime; refining this criterion would help improve future hazard assessments.

### 470  7  Conclusions

In 2019, the ice-dammed Lac des Faverges, located on Glacier de la Plaine Morte in the Swiss Alps, partially drained through an artificial supraglacial channel, constructed in order to mitigate the hazard posed by this lake. This unique setting was used to acquire a comprehensive dataset describing the evolution of the lake level, channel discharge, channel incision, and channel-water temperature during drainage. It is probably one of the most comprehensive such datasets currently available.

The field measurements were used to characterize the hydraulics and thermodynamics of the supraglacial channel, quantifying, among other parameters, the friction factor and the Nusselt number. The observed Darcy-Weisbach friction factors range between 0.17 and 0.48, with a mean value of 0.30 which is close to what other studies found and to what modelling studies





used so far. However, the large spread is indicative for the importance of considering the friction factor as a stochastic variable, instead of as a constant.

The heat transfer between water and channel wall, responsible for channel incision and quantified by the Nusselt number, was determined using two distinct methods: the *melt rate method* and the *spatial cooling-rate method*. The results of the two methods agree, but as for the hydraulic friction factor, a large spread is found, indicating that the Nusselt number should also be treated as a stochastic variable in modelling studies. The Nusselt numbers derived from the often-used empirical Dittus-Boelter equation are significantly different to ours. More precisely, the dependence of the Nusselt number on the Reynolds number is

less pronounced than previously reported (e.g. Clarke, 2003; Lunardini et al., 1986; Vincent et al., 2010) or in the commonly used empirical Gnielinski correlation. We identify the heat-transfer rate as one of the key processes to be investigated by future studies, since it defines the supraglacial channel incision rate and thus the discharge and lake drainage stability. For this to be successful, an increase in the representativeness and accuracy of water temperature measurements would be needed.

    The modelling of supraglacial lake drainages will likely remain afflicted by large uncertainties. In line with previous work,

our study shows that some key hydraulic and thermodynamic parameters are only weakly constrained. This translates into large model uncertainties which, in turn means that model-based hazard assessments will need to continue allowing for large margins of errors.

## Appendix A: Calculation of $Nu$ as a function of the e-folding length $x_0$

We derive the spatial temperature profile along the channel (Eq. (11) and Eq. (12)) using the energy conservation equation

$$\frac{\partial E}{\partial t} + \frac{\partial vE}{\partial x} = M,\tag{A1}$$

where $E = \rho_w c_w T_w A$ is the energy density of the water per unit channel length ($\mathrm{J\,m^{-1}}$), $x$ the distance (m) along channel-flow path, $v$ the stream flow velocity ($\mathrm{m\,s^{-1}}$), and $M$ is a source term ($\mathrm{W\,m^{-1}}$). The source is expressed in terms of $q$, the heat flux ($\mathrm{Wm^{-2}}$) from the water into the ice

$$M = qP_w.\tag{A2}$$

We thus assume that the only relevant heat source stems from ice melt at the channel wall and that both heat exchange at the ice-air interface and heat production due to potential energy dissipation can be neglected. This is justified as the those two heat sources on the order of $100\,\mathrm{Wm^{-1}}$, which is two orders of magnitude smaller than $M$. We write $q$ as

$$q = h_t\Delta T,\tag{A3}$$

where $h_t$ is the convective heat transfer coefficient (the conductive heat transfer can be neglected) and $\Delta T$ is the temperature

difference between water and ice. Note that since the ice temperature is 0°C, $\Delta T = T_w$. Assuming a steady state, i.e. $\frac{\partial E}{\partial t} = 0$, using Eq. (A3), and expressing $E$ in terms of $T_w$, the Eq. (A1) becomes

$$Q\,c_w\rho_w\frac{\partial T_w}{\partial x} = h_t T_w P_w,\tag{A4}$$





which uses the approximation $\frac{\partial Q}{\partial x} \approx 0$. This equation can be integrated to give

$$T_w = T_0\, \mathrm{e}^{-\frac{x}{x_0}}, \tag{A5}$$

with

$$x_0 = c_w \rho_w \frac{Q}{h_t P_w} = \frac{c_w \rho_w \lambda}{k_w} \frac{Q}{Nu\, P_w}, \tag{A6}$$

where the second equality follows from Eq. (8).

*Code and data availability.*    The raw data, the code to process the raw data, and the results are available at https://people.ee.ethz.ch/~werderm/
pm-data/. The data, code and results will be later referenced under the same format with a DOI on ETHZ Research Collection (https:
//www.research-collection.ethz.ch/) after acceptation for publication.

*Author contributions.*    CO conducted the field campaign, did the data analysis, produced the figures, and wrote the paper. MW designed the
field campaign, participated in it, developed the general methodological aspect of the study, and co-wrote the paper. MH provided inputs on
the Methods, helped with fieldwork, and gave feedback on the paper. IK and DH provided information on the channel construction, measured
and provided data, and gave feedback on the paper. DF did the overall supervision, gave feedback on the paper, and acquired funding.

*Competing interests.*    The authors declare that they have no conflict of interest.

*Acknowledgements.* The field campaign was partly funded by the WSL internal project *Glacier lake outburst floods – a full-
scale experiment (GLOFFEE)* . The additional support by La Prairie and the ETH Zurich Foundation are kindly acknowledged.
The authors thanks all the helpers who took part in the field campaigns: L. Geibel, E. Hodel, J. Klahold, S. Förster, F. Lindner,
A. Sergent, and F. Walter. We appreciated the collaboration with Geotest AG for field logistics.




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
