# Peer review of "Drainage of an ice-dammed lake through a supraglacial stream: hydraulics and thermodynamics"

_The Cryosphere, 2021_

## Author Response (AR1)

**We would like to thank the reviewer 1 for the corrections proposed. Below we have answered all reviewer comments (RC1) and state how we address them in the revised manuscript (proposed new text in quotation mark and in italic).**

**RC1:** Am I right that the ice-dammed lake is located at the margin of the ablation zone of the Glacier de la Plaine Morte where 4--5 m thick winter snow was removed by snowcats and the ice channel then dug into solid ice (cut by an excavator)? Hence, the channel wall was made of solid ice impermeable to water.

**Authors**: This is exact. The Equilibrum-Line Altitude of Glacier de la Plaine Morte has been above the glacier highest elevation for many years, and the accumulation/firn area had been depleted since a while. We added to the text at L105:

*"In a first stage, the 4-5 m deep snow cover had to be removed by snowcats. In a second stage, the solid and impermeable ice was cut and removed by an excavat."*

**RC1:** Line 16. Has drainage through englacial conduits from an ice-dammed glacial lake been reported?

 **Authors**: Yes. Roberts (2005) presents the different types of drainage in his Table 1 (in relation with his Figure 1). He mentions englacial drainage as "intraglacial drainage" in Type 2 and Type 5 (Table 1). We thus left the references as such in the text.

**RC1:** Line 19/23. Pre-existing veins rather than cracks?

 **Authors**: "Veins" is indeed more appropriate than "cracks" in this context (also in respect of Nye 1976 paper). Modification done.

**RC1:** Line 57. "Purely driven by physics" i. e. by physical processes or hydraulic processes.

 **Authors**: We replaced "by physics" by "*by ice physical and hydraulical processes*" to be clearer.

**RC1:** Line 97. Does the constructed supraglacial channel connect the lake to an englacial moulin?

 **Authors**: The connection between the supraglacial channel and the lake is not really a moulin. To our understanding, a moulin route the supraglacial water to the subglacial system. Instead, we call this connection an englacial syphon, since it connects horizontally the lake and the canyon-channel system (see Fig. 1b), without interaction with the subglacial system. Note that there is indeed a moulin called Moulin West at the end of the supraglacial construction (see Fig. 1a), and also one in the middle of the supraglacial construction, in the so-called "micro tunnel" (Fig. 1c), where the channel water entered the glacier in 2019.

**RC1:** Line 239  0.5 m.

**Authors**: Modification done.

**We would like to thank the reviewer 2 for the very detailed and constructive review. Below we have answered all reviewer comments (RC2) and state how we address them in the revised manuscript (proposed new text in quotation marks and italic).**

Most importantly we have updated the manuscript in the following points:

- included information on the lake volume uncertainty (see details below),
- expanded the description on the model used to calculate the water input in the lake basin,
- added an Appendix B to present all salt dilution experiments used to calculate the hydraulics and thermodynamics parameters.
- added information on the 2020 and 2021 lake drainage event and the role of the supra glacial channel in section Discussion,
- corrected all typography mistakes and followed the re-phrasing proposed.

**RC2:** L21: add the Mayer&Schuler reference (Breaching of an ice dam, Annals of Glaciology, 2005) to complete the list of studies documenting type iii) drainage

**Authors:** Addition done.

**RC2:** L114: can you provide an uncertainty measure for the lake volume?

**Authors:** Thanks for this remark. We agree that this is an important information and we have now added it. The actual uncertainty is relatively difficult to constrain and is mainly affected by three uncertainty sources: (1) The lake volume on 10 July 2019 was calculated using a DEM acquired by Swisstopo on 28 August 2018. Consequently, we expect some additional melting happening during September 2018, which results in slight, difficult-to-quantify changes of the lake bathymetry. Melting could also have occurred due to direct contact between the lake-damming ice and the lake water (heat transfer from water to ice) in May/June/July 2019, although we note that the lake floor was partly still covered with winter snow when the filling set in, meaning that the ice determining the actual lake bathymetry was protected from melt. Note that roughly half of the lake floor lies on ice (Figure 1.b), and is therefore affected by this ice-melt related uncertainty. To constrain the lake volume on 10 July 2019, we calculated the lake volume using the DEM acquired in September 2019 as an upper limit of uncertainty ($1.59 \times 10^6$ m$^3$), and the lake volume using the DEM acquired in late August 2018 as a lower limit of uncertainty ($1.38 \times 10^6$ m$^3$), both for the same lake level elevation corresponding to 10 July 2019. We took the average of the two volume to determine the maximum volume value, with the two extremas constituting the uncertainty range. (2) The used Swisstopo DEM is also affected by uncertainties. These are specified as +/-2m in the vertical at the pixel scale. We however can assume this pixel-based uncertainty to be spatially uncorrelated. Thus, the overall uncertainty in computed lake volume is reduced with $sigma/\sqrt{N}$, where $sigma$ is the DEM pixel-based uncertainty for each pixel (+/-2m) and $N$ the number of pixels (several thousands), resulting in an uncertainty of +-0.01 m. (3) The maximum lake level elevation, measured by differential GPS, is also affected by uncertainties. This uncertainty is estimated to be in the order of a few millimetres, and is thus negligible compared to (1).

In the revised version of the text, we will address the above points by adding the following paragraph:

*"The main uncertainty in our estimate of lake volume is the poorly constrained ice-surface melt occurring in the lake basin between late August 2018 (date of DEM acquisition) and July 2019. This results in bare-ice melting in autumn 2018, and in bare-ice melting due to heat transfer from water to glacier-ice before the lake drainage. Since these melt processes are not quantified in the lake basin, we constrained the lake volume using DEMs from August 2018 ($1.38x10^6$ $m^3$) and September 2019 ($1.59x10^6$ $m^3$) as a lower and upper bound, respectively. We determine the volume to be the average of the two bounds, i.e. $1.49$ $x10^6$ $m^3$ +/- $0.11$ $x10^6$ $m^3$. Note, for the subsequent calculation of lake outflow the bathymetry of the lake is required. For this, we use the 2018 DEM because ice-melt between 28 August 2018 to 10 July 2019 is expected to be significantly smaller than between 10 July 2019 to 3 September 2019."*

**RC2:** L127: hydraulic slope (dimensionless, expressed as water head drop per horizontal channel length)

**Authors:** Modification done

**RC2:** L128: …the velocity averaged over the cross-section…

**Authors:** Modification done

**RC2:** L154: …were marked with stakes…(plural)

**Authors**: Modification done

**RC2:** L156: GPS accuracy: in the horizontal or in the vertical?

**Authors**: In the vertical. We thus explicitly wrote *"vertical accuracy of…"*

**RC2:** L157ff: CTD description: what are the specifications for the conductivity probe?

**Authors:** The measurement range of conductivity is [0-0.002] S/m. The accuracy is 2.5% of the range, so $5 \times 10^{-5}$ S/m. We added that to the text.

**RC2:** L161: the stated uncertainty of the water level measurements is in the same range as the readings presented in Fig 5. Isn't this a major problem? how significant are the recorded variations?

**Authors**: We made a mistake on the specifications of the pressure measurements. The standard deviation is actually 0.05% of the measurement range. The latter is 10 m in our case, resulting in an water level uncertainty of 5 mm, and not 0.2 m, as erroneously stated before. We modified the text accordingly.

**RC2:** L169: uncertainty (instead of error)?

**Authors**: Yes. Modification done.

**RC2:** L170: please describe how you picture the formation of these melt-imprints and how this relates to diurnal discharge variations.

**Authors**: We have a hypothesis that we formulated as follows:

Revised text: *"We suppose that the deeper incised sections of the melt-imprints form during the afternoon, when relatively high water temperature and discharge yield to significant melt on the channel walls. Conversely, decreasing discharge and water stage during the night yields to less side-way melt on the wall section which then emerges from the water, thus producing the less deeply incised sections of the melt-imprints."*

We think also that further research is needed to fully understand their formation.

**RC2:** L179/180: either 'salt dilution' or 'tracer dilution', 'salt tracer dilution' is redundant

**Authors**: We replaced "salt tracer dilution" by *"salt dilution"*.

**RC2:** L180: referring to this method is ambiguous, there exits more than one dilution method, (continuous vs instantaneous injection)

**Authors:** We clarified it by mentioning "salt dilution" method instead of "salt trace dilution" (see comment above). We believe that this is now less ambiguous, since this specific salt dilution method that we use is explained in the book of Hubbard and Glasser.

Revised text: *"Channel discharge $Q_i$ was measured using the salt dilution method (Hubbard and Glasser, 2005). We carried out 33 instantaneous salt injections at station P5 on 12 different days during the campaign."*

**RC2:** L188: …which was the case for all presented measurements. How do you know this?

**Authors:** When the sensors were not at the bottom of the channel, we could guess it from the water stage time series that were noisy and close to the atmospheric pressure. In that case we discarded the data. Certainty about the sensors being at the bottom also came from visual inspections, and thanks to the weight coupled with the sensors, which ensured that the water pressure corresponds to the maximum water stage. We added this information.

Revised text: *"The measurements rely on the pressure transducer sinking to the bottom of the channel. This is ensured to be the case for all presented measurements, thanks to repeated visual inspection during field visits and because pressure transducers were weighted. When pressures transducers were not at the bottom of the channel, times series were noisy and close to the atmospheric pressure value, and we discarded the data."*

**RC2:** L191: clarify: correcting a shift is not the same as filling a gap

**Authors**: Indeed, we removed "i.e. the data gaps were removed".

**RC2:** L198: The model has been calibrated using the seasonal mass balance data collected by…

**Authors**: Modification done.

**RC2:** L200ff: please add information to complete model description: what is the period over which the model has been applied? State source and location of data used for model forcing. How did you deal with precipitation? Simply using the meteorological records or some adjustments?

**Authors:** The following information was added: *"We applied the model from September 2018 to September 2019 with a daily resolution to the watershed of the lake, and used it to estimate $Q_{in}$ consisting of snowmelt from the glacierized and ice-free portion of the basin, bare-ice melt and liquid precipitation. The distributed mass balance model (e.g. Huss et al., 2021) was driven with meteorological observations from Montana (9 km from the study site) and both melt factors as well as a precipitation correction factor have been calibrated to match seasonal mass balance observations on Plaine Morte in 2021."*

**RC2:** L210ff: the description of salt dilution measurements should be moved from 'data processing' to 'field measurements' (sect 4.1)

**Authors:** We agree that conductivity field measurements are best placed in the "Field measurements" section (Sec. 4.1). In sub-section 4.1.2 entitled "Water pressure, temperature and conductivity in the channel" we presented the measurements of conductivity in the channel. This is where the sensors specificity on conductivity is presented, together with the location of salt injection and readings. However, we think that the description of the data processing of conductivity measurements should stay in the "Data processing" Section (in the sub section 4.2.2 "Hydraulic"). This is because it only explains the processing steps made in the laboratory/office (e.g. sensors calibration, conversion of conductivity readings into salt concentration and then into discharge), and not the ones in the field.

**RC2:** L212: were the concentration standards prepared using in situ water or de-ionized water? In the second case, how did you deal with naturally occurring background concentrations?

**Authors:** The calibration solution was made in de-ionized water. The background concentration in the field was removed. The background level was easy to identify because its value is a constant plateau before and after the (short) conductivity peak. We added to the text: *"First, the natural background level of conductivity at these stations was removed for each injection, and the readings were converted…"*

**RC2:** L214: …were integrated over the time of the tracer passage, for each injection…

**Authors:** Modification done.

**RC2:** L215ff: how many data samples have been used to establish the rating curve?

**Authors:** We used 12 data samples. This information is now given through the new Table in Appendix B.

**RC2:** L225: wording: a gap in water level records cannot be filled with discharge.

**Authors**: Indeed, now reads: *"A data gap in the channel's water stage time series between 13 and 24 July 2019 was filled using values based on daily lake discharge calculated using Eq. (3)."*

**RC2:** L228: hydraulic slope or hydraulic gradient, use either or and be consistent throughout the paper, avoid synonymous expressions.

**Authors**:  We decided to use "hydraulic slope" in the following.

**RC2:** L235: …is the wetted cross-sectional area…

**Authors:** Modification done.

**RC2:** L250ff: since you describe Nu and Pr, do it for Re as well, what does it characterize?

**Authors**: We added the definition bellow:

Revised text: *"The Reynolds number Re (dimensionless) is the ratio of inertial forces to viscous forces within a fluid, and quantify the turbulent flow."*

**RC2:** L270ff: make this clear: there exist two parametrizations for Nu, in addition, you use two further methods to determine it. In the present text, this is confusing.

**Authors**: To clarify our intentions, we modified the text into *"In addition to these two empirical relations, we present below two alternative methods to calculate Nu directly from our measurements (i.e. without using a parametrisation)."*

**RC2:** It would be very helpful to have a table listing all salt dilution measurements and their results (date, time, Q, v, S, used for rating curve y/n), but this may be material for the appendix.

**Authors:** We agree that this information could be relevant for the readers. However, the information requested is too much to be fitted into a  single table. We therefore decided to create a table which only present the measurements done between stations P5 and P3, which is the section of interest for our calculations of hydraulics and thermodynamics. We also added the water stage measurements used to construct the rating curve. This table is now referenced in section " 4.2.2 Hydraulics". The others values at P2 and P1 are found in the section "Data and code availability".

**RC2:** L300: …of the latter compared to previous years.

**Authors:** Modification done.

**RC2:** L300: the increasing trends of both…are visible (plural)

**Authors:** Modification done

**RC2:** L302: In warmer years, the date….

**Authors:** Modification done

**RC2:** L303: …depletion of the winter snow cover…

**Authors:** Modification done

**RC2:** L307/8: We distinguish four different phases.

**Authors:** Modification done

**RC2:** L316: The channel bottom elevation and its evolution with time …   The incision shows…

**Authors:** Modification done.

**RC2:** L320: In contrast, the higher slope…led to more turbulent water flow and subsequent formation of step-pools

**Authors:** Modification done.

**RC2:** L321: Note that meandering did not occur…

**Authors:** Modification done.

**RC2:** Fig2: please use a consistent formatting of timestamps on the x-axis. Here the format is DDMMM but in Figs 5,6 and 8 it is MM/DD

**Authors:** We thank for the comment. To ensure consistency, we changed the format of Fig 2 to MM/DD.

**RC2:** Fig3: vertical axis should be simply labeled 'Elevation'

**Authors**: We changed the label to "Elevation (m a.s.l)" as proposed, but added *"Channel bottom elevation"* as a legend title to avoid confusion and to clarify what the lines represent.

**RC2:** Fig3: the color code for the lake level seems not to correspond to that of the channel bottom measurements. As is, the lake level is always lower than the channel bottom (for all dates), and since this is a subaerial lake, it is impossible that it would drain through the channel.

**Authors:** This was a mistake from our side. We modified the color scale as suggested. Note that there is no lake level for the 4[th] September since the lake was empty at this date.

**RC2:** Fig6: use larger symbols to denote the measurement points, at the present size, the different symbols are difficult to discern.

**Authors:** Modification done.

**RC2:** Fig6a: vertical axis should be simply labeled *'Elevation'*

**Authors:** Modification done.

**RC2:** L373: It is noteworthy that our results…

**Authors:** Modification done.

**RC2:** L378: We collected an extensive…

**Authors:** Modification done.

**RC2:** L406: is an indication for slush-ice ….but also for the influence of form friction (as opposed to skin friction) in a complex 3d channel geometry

**Authors:** Modification done.

**RC2:** L422: …than our values and those of …

**Authors:** We now write: *"Our results (Fig. 8 and 9) are in-between the predictions of the Dittus-Boelter equation (Eq. (9)) using the parameters from Vincent et al. (2010), which are higher than our values, and those of Lunardini et al. (1986), which are lower than our values."*

**RC2:** L423: …of these two studies…

**Authors:** Modification done.

**RC2:** L436: …the basis for estimating Nu in our …

**Authors:** Modification done.

**RC2:** L448: …was considered by Jarosch and Gudmundsson (2012) who suggest…

**Authors:** Modification done.

**RC2:** L451: …but extending the study of… could shed light on this issue.

**Authors:** Modification done.

**RC2:** L455: state the volumes!

**Authors:** Modification done.

**RC2:** L458: Still, construction costs were lower than the damage…

**Authors:** Modification done.

**RC2:** Sec 6.4: what are the perspectives for the future? Will the existing channel close are may it be re-used in subsequent years (then at a fraction of the costs!) what happened to the lake in 2020?

**Authors:** No dedicated field campaigns were conducted in 2020 and 2021 but we added some qualitative information for these years to the Discussion.

 *"The artificial channel remained active throughout the summer of 2020 and it was effective in limiting the lake volume and thus the hazard emerging from it. In terms of operations, the substantial amounts of winter snow blown into the channel proved to be challenging, as they*

*formed an intermediate blockage. In early August 2020, the winter-snow was partially removed by an excavator, and the remaining snow blockage was eroded in a slush-flow like event, after which the lake drained partially through the supraglacial channel (i.e. corresponding to Phase I in Figure 6), and partially subglacially (i.e. corresponding to Phase IV in Figure 6). Since the slush-flow like event was relatively difficult to control, additional artificial measures were taken for 2021. These aimed at activating the channel's water flow underneath the extensive snow cover that rebuilds during winter. However, in 2021 the lake drained subglacially when only half full and before lake water could flow through the channel; the lake outlet was situated in the "canyon" region (Fig. 1b). The drainage occurred in late July 2021 which was early relative to the still extensive snow cover and to the low filling level."*

**RC2:** L478: However, the large spread found in our study suggests considering the friction factor…

**Authors:** Modification done.

**RC2:** L489: …afflicted with…

**Authors:** Modification done.

**RC2:** L491: …will have to allow for large uncertainties.

**Authors:** Modification done.

**RC2:** L496: A is the cross-sectional area (you used the symbol S in eq 6)?

**Authors:** Correct, this was a mistake from our side. *S* is the wetted cross section area.

**RC2:** L500: …the only relevant heat source is negative and stems from the consumption of energy related to ice melt…

**Authors:** Modification done.

**RC2:** L501: This can be justified, as these two sources are on the order…

**Authors:** Modification done.

**RC2:** L508: …which uses the assumption dQ/dx =0

**Authors:** Modification done.

---

## Author Response (AR2)

Dear Editor,

Thanks for the additional corrections suggested. Here a summary of the last corrections we implemented before sending our final paper version:

**Editor:** line 99: elevation
**Authors:** Modification done

**Editor**: line 101: To my experience melt season is the more familiar term.
**Author**s: Modification done

**Editor**: line 120: A slope of 0.34% corresponds to an elevation difference of only 1.8 m along the 540 m long channel. Is that correct? it seems more than that in Fig. 1d?
**Authors**: Indeed, we probably did this mistake by taking only the shorter upperpart of the channel in the calculation. The slope between P5 and P1 is 0.72% (3.33 m of elevation for 464.4 m of distance). We implemented this new slope value.

**Editor**: line 134: The more common units for thermal conductivity are W m-1 K-1
**Authors:** We did the modification, and we also added "K" instead of "°C" for the other units to respect the S.I standard (example: the specific heat capacity of water) and make the text more consistent. Note that we kept "°C" for absolute measurement of temperature.

**Editor**: line 165: record
**Authors**: Modification done

**Editor**: line 359: indicates
**Authors**: Modification done